# In-Field Automatic Detection of Grape Bunches under a Totally Uncontrolled Environment

**DOI:** 10.3390/s21113908

**Published:** 2021-06-05

**Authors:** Luca Ghiani, Alberto Sassu, Francesca Palumbo, Luca Mercenaro, Filippo Gambella

**Affiliations:** 1Department of Agricultural Sciences, University of Sassari, Viale Italia 39 a, 07100 Sassari, Italy; lghiani@uniss.it (L.G.); asassu@uniss.it (A.S.); mercenaro@uniss.it (L.M.); 2Intelligent System DEsign and Applications (IDEA) Group, Department of Chemistry and Pharmacy, University of Sassari, Via Muroni 23/A, 07100 Sassari, Italy; fpalumbo@uniss.it

**Keywords:** deep learning, grape detection, object detection, precision agriculture, precision viticulture

## Abstract

An early estimation of the exact number of fruits, flowers, and trees helps farmers to make better decisions on cultivation practices, plant disease prevention, and the size of harvest labor force. The current practice of yield estimation based on manual counting of fruits or flowers by workers is a time consuming and expensive process and it is not feasible for large fields. Automatic yield estimation based on robotic agriculture provides a viable solution in this regard. In a typical image classification process, the task is not only to specify the presence or absence of a given object on a specific location, while counting how many objects are present in the scene. The success of these tasks largely depends on the availability of a large amount of training samples. This paper presents a detector of bunches of one fruit, grape, based on a deep convolutional neural network trained to detect vine bunches directly on the field. Experimental results show a 91% mean Average Precision.

## 1. Introduction

Precision agriculture evaluates spatial and temporal variability of field data through automatic collection and digitization of extensive information databases. Different types of sensors are applied to develop high-efficiency approaches to optimize input use, maximize crop production, reduce wastes, guarantee environmental sustainability, and obtain economic benefits [1,2,3,4]. These specific approaches apply to viticulture in terms of efficient use of inputs, such as fertilizers, water, chemicals, or organic products [5,6]. In this context, improving crop protection, the use of machinery and labor for harvesting, pruning, or other crop management operations focuses on improving the efficiency of each plot within the vineyard [7,8,9,10,11]. Vineyards are characterized by high spatial and temporal heterogeneity and are influenced by pedo-morphological characteristics, climate, phenology, and cropping practices [12]. These variables can influence grape yields and quality, and their prediction is the main goal of precision viticulture. Farmers can be encouraged to pursue the economic benefits and achieve the desired oenological results by the latest technologies combined with decision support techniques [13,14,15].

Emerging viticulture technologies are not fully developed, and several challenges still need to be addressed. While much of the work is currently promising, much effort is required to the so-called “vineyard of the future”. Viticulturists may therefore get advantage of modern tools to monitor and tailor the management of their vineyards. Useful data in vineyard management include automatic knowledge of the leaf area, fruit harvesting, yield estimation, evaluation of grape quality, and grapevine cultivar identification [16,17].

Yield estimation is of critical importance in the wine industry. Traditionally, yield forecasts were based on manual counting of grapevine numbers, bunch numbers per vine, and include manual and destructive sampling of bunches to determine their weights, berry size, and number.

When computer vision [18] and machine learning [19] are considered, object detection [20] represents a technique that deals with the detection of one or more categories of objects via the digitization of a given image or video. Object detection tasks can be roughly split into the object localization (where each object is located within the image) and object classification (which category each object belongs to). The location of the bounding box around each detected object will be returned in pixels as the x and y coordinates of the top left corner, and the width and height of the box. In this work, we only look for objects belonging to the grape class.

As in many other applications of machine learning, in the last decade, deep learning [21] methods proved to be among the most effective in object detection [22,23]. Many different techniques have been implemented starting from R-CNN (Region Based Convolutional Neural Networks) [24], Fast R-CNN [25], and Faster R-CNN [26] up to, among many others, YOLO (You Only Look Once) [27] and Mask R-CNN [28].

Many of these techniques have been successfully applied in the agricultural field. The aim of the work of Sa et al. [29] was the building of a fruit detection system. Using transfer learning and fine tuning techniques they were able to train a multi-modal Faster R-CNN model with a really limited amount of images. They combined RGB and NIR (Near InfraRed) information building a reliable multi-modal system. The comparison with a Conditional Random Field with hand-crafted features method previously presented by the same team proved the validity of the approach. Bresilla et al. [30] trained a YOLO convolutional network for fruit detection and localization in images of apple and pear trees. Preliminary results were improved by some network modification, dataset augmentation, and also the generation of synthetic images. The network was first trained to detect apples using apple trees images. The trained network was then “fine-tuned” with pear tree images to also detect pears. To estimate the biovolume of olive trees, Safonova et al. [31] used deep learning instance segmentation methods. They analyzed RGB images and two well-known normalized difference vegetation indexes. Several Mask R-CNN-based models were used for the segmentation of olive tree crown and shadow to estimate the biovolume of individual trees. Fuentes et al. [32] proposed a robust deep-learning-based detector for real-time tomato diseases and pests recognition. Several experiments were conducted with an in-depth analysis of various deep learning architectures and feature extractors. Accuracy was further increased by data augmentation techniques and the system was able to effectively recognize nine different types of diseases and pests. Picon et al. [33] presented several crop disease classification methods for mobile devices (Android, iOS, and Windows Phones) using a Deep Residual Neural Network with 50 layers and 224 × 224 input image size. They first extended an already existing dataset collecting leaves images in Spain and Germany. The leaves were labeled as affected by Rust, Septoria, Tan Spot, or Healthy. Three different kinds of inputs were provided to a neural network: the resized full image, a leaf mask crop, and a superpixel based tile. Several data augmentation techniques were applied and the training phase was repeated adding an artificial background to the images. Experimental results proved to be interesting with significant increases caused by the super pixel segmentation, the artificial background, and the image augmentation.

In order to detect bunches of grapes or single berries, several methods have been proposed. Reis et al. [34] were able to detect red and white grapes, experimentally selecting a few intervals of Red, Green, Blue (RGB) values by trial and error, collecting images during the night to limit light/brightness variations. After a sequence of iterations of the morphological dilation, the bunch regions were located and measured. Diago et al. [35] automatically estimated the number of flowers per inflorescence. The images of the inflorescences, with a uniform background of black color, were first converted from the RGB to the CIELAB color space (CIE L*a*b, where CIE stands for International Commission on Illumination in French), then segmented using thresholding based on histogram values. The elimination of local peaks (lower than a threshold) and a final post-processing filtering allowed to find and identify the brighter points corresponding to the flowers. An automatic system for shoot detection and yield estimation has been proposed by Liu et al. [36]. Images are converted from the RGB to the L*a*b color space and an Otsu thresholding technique [37] is used for the first segmentation step. An unsupervised feature selection followed by an unsupervised shoot classification using the K-means clustering algorithm leads to the shoot identification. Diago et al. [38] proposed a methodology to characterize the grapevine canopy and assess leaf area and yield through RGB images. They used the Mahalanobis distance to classify leaves (young or old), wood, grapes, and background. Font et al. [39] acquired images at night under controlled artificial illumination to simplify the grape segmentation procedure. They analyzed both the RGB and the Hue, Saturation, and Value (HSV) color spaces and segmented the images with five different methods: thresholding with the Otsu method followed by a sequence of morphological filtering; Mahalanobis distance between the three-dimensional color intensities; a Bayesian classifier; a Linear Color Model; a three-dimensional color-intensity histogram.

A methodology for segmenting inflorescence grapevine flowers was presented by Aquino et al. [40]. They applied some morphological operators to the images in the HSV color spaces and a top-hat transformation to emphasize bright details. After binarization and pyramidal decomposition, the regional peak corresponding to the inflorescence was found. An automated image analysis framework for berry size determination was proposed by Roscher et al. [41]. Working in the YIQ color space and after the detection of berry candidates with the circular Hough transform, they extracted several features from the image patches around the detected circles. Berry diameters were measured after using conditional random field to classify those patches as berry or non-berry. Another berry detection method using images converted to the CIELAB color space was proposed by Aquino et al. [42]. Images were acquired with dark cardboard placed behind the cluster. After an Otsu thresholding and some filtering, berries candidates were selected by finding regional maxima of illumination, and then six descriptors were extracted, and false positives were discarded using two different supervised-learning classifiers: Neural Network and Support Vector Machine. Liu and Whitty [43] eliminated irrelevant regions in the image by thresholding the H and V channels in the HSV color space obtaining potential bunch areas and reduced the noise by applying several morphological operations. The resulting bunches in 80 images were manually labeled as true or false and 54 different measures from RGB, HSV, and L*a*b color spaces were extracted. After applying the ReliefF algorithm and a sequential feature selection to reduce the feature dimensions, the SVM was used to train the system. Nuske et al. [44] predicted yields in vineyards through cameras and illumination mounted on a vehicle. They detect potential berry locations using a Radial Symmetry Transform and an Invariant Maximal Detector, then they use Gabor filters, a SIFT descriptor, and a Fast Retinal Descriptor to classify the detected points as grapes or not-grapes through a randomized KD-forest. To avoid double-counting of grapes between consecutive images, the grape locations were registered. A sequence of calibration measurements allows the team to predict yields with remarkable precision. That work was continued by Mirbod et al. [45] that used two algorithms (Angular invariant maximal detector and Sum of gradient estimator) for berry diameter estimation. Coviello et al. [46] introduced the Grape Berry Counting Network (GBCNet). It belongs to the family of Dilated CNNs and it is composed by ten pre-trained convolutional layers for feature extraction and by a dilated CNN for density map generation. The authors were able to estimate the number of berries in the image achieving good performances on two datasets, one with seven different varieties and one with only one variety. Finally, a more comprehensive review of computer vision, image processing, and machine learning techniques in viticulture has been proposed by Seng et al. [47].

Three main limitations characterize many of the works summarized in this section: the detection process is not fully automated, it is usually based on a limited amount of data (dozens or hundreds of images), and it is also based on a limited amount of grape variety (in most of the cases no more than two). Therefore, a method applied on images acquired in a vineyard under specific conditions may not work as well in another vineyard or may not even work in the same vineyard as some of those conditions change.

In this paper, we try to overcome each of these problems. As a matter of fact, the aim of this work was the development of a grape detector able to analyze images automatically acquired by a vehicle moving in a generic vineyard (located in an unspecified geographical area with an unspecified grape variety). Due to those issues, in Table 1, the main characteristics of the grape detection method proposed in this work are compared with those presented in Section 1, focusing on the detection process, the data set and the number of grape varieties. The detector, based on an R-CNN (Region Convolutional Neural Network), was trained and tested on the GrapeCS-ML dataset containing more than 2000 images of much different varieties described in the next section. We also used an internal dataset to further test the framework on different grape varieties and under different environmental conditions.

## 2. Materials and Methods

In this section, we will fully describe the proposed methodology summarized in Figure 1. The GrapeCS-ML dataset was labeled and divided in train, validation, and test subsets. Augmentation techniques [48] were applied to the training subset. A pre-trained Mask R-CNN framework was fine-tuned using the train and validation subsets, and from the trained network and the test subset were obtained the experimental results.

### 2.1. Dataset

The main difficulty in applying machine learning techniques in the agronomic field is the availability of useful data for training and testing. In 2018 the Charles Sturt University released the freely downloadable (as a zip file) GrapeCS-ML dataset [47], containing more than 2000 images of 15 grape varieties at different stages of development and collected in three Australian vineyards. The images are divided into five subsets:Set 1: *Merlot* cv. bunches, taken in seven rounds from the period January to April 2017;Set 2: Designed for research on berry and bunch volume and color as the grapes mature, featuring Merlot, Cabernet Sauvignon, Saint Macaire, Flame Seedless, Viognier, Ruby Seedless, Riesling, Muscat Hamburg, Purple Cornichon, Sultana, Sauvignon Blanc, and Chardonnay cvs;Set 3: Subsets for two cultivars (*Cabernet Sauvignon* and *Shiraz*) taken at dates close to maturity;Set 4: Subsets of images for two cultivars (*Pinot Noir* and *Merlot*) taken at dates close to maturity, with the focus on the color changes with the onset of ripening;Set 5: *Sauvignon Blanc* cv. bunches taken on three different dates. Each image also contains a hand-segmented region defining the boundaries of the grape bunch to serve as the ground truth for evaluating computer vision techniques such as image segmentation.

Although several subfolders contain some data such as the grape variety and the date of acquisition, a meaningful information is missing: the ground truth, i.e., the position of the bunches inside the different images. Therefore, we hand-drew the smallest Bounding Boxes around every bunch of grapes for each image. We used the “Image Labeler” app (Figure 2) available within Matlab. As shown in the Figure, the app enables the user to define a set of class labels (in our case just one class named “grape”) to draw a rectangle that is the Region of Interest (RoI) around each selected object and to label that ground truth as belonging to one of the previously defined classes.

A color reference or a volume reference is present in most of the images (a few examples are shown in Figure 3) but we chose to ignore this kind of information in order to obtain a fully automated detection process.

During the last 15 years, thousands of digital images of bunches were collected at the Department of Agricultural Sciences, University of Sassari (a few examples are presented in Figure 4).

While all the GrapeCS-ML images of different grape varieties were collected in Australian vineyards, the ones in our dataset were collected all around in Sardinia Island (Italy), literally on the other side of the world. The number of available images were in the thousands and they were acquired all around several Sardinian vineyards. Some contained the entire vineyard, others in perspective the space between two rows or an entire row imaged from one end. The purpose of our work was to train a detector able to analyze images automatically acquired by a vehicle moving between the vine rows. Therefore, we only selected photos acquired between the rows at a distance of about one meter from the leaf wall. A total of 451 images were selected to further test the trained network. It is worth emphasizing the importance of testing the system on a dataset that contains images like those we will work on. Moreover, it would be even more important to ascertain the ability of the system to provide good detection results on images very different from those present in the training set. In fact, while in the former case, we would have a well performing detector on a specific vineyard, in the latter we would have a “universal” detector able to work anywhere.

### 2.2. Mask R-CNN Framework for Grape Detection

Given its performance on several well-known object detection benchmark datasets [22,23], we have chosen to train our system with the Mask R-CNN method [28]. The Python implementation used in this work is freely downloadable from https://github.com/matterport/Mask_RCNN (accessed on 3 June 2021) [49].

The Mask R-CNN framework (Figure 5) segmentation is an extension of Faster R-CNN, and it adopts a two-stage procedure.

The first stage is called Region Proposal Network (RPN) and is a fully convolutional network. The RPN can be trained to predict region proposals at different scales and aspect ratios; therefore, it is used to estimate the position of bounding boxes. The second stage corrects the RoI misalignments in the RoIAlign layer and then performs in parallel a classification, a bounding box regression, and extracts a binary mask in order to output the bounding boxes and the segmentation masks of the selected object [28]. In this work, we only trained the system to extract the bounding boxes values while ignoring the segmentation.

### 2.3. Training Procedure

It is well-known that deep learning training process requires a huge number of samples, hundreds of thousands, or even millions. In addition, training a model from scratch is tremendously expensive in terms of required computational power but also in terms of processing time. Luckily, the availability of a pre-trained model allows the execution of the so-called “fine-tuning”. In the fine-tuning process, a model trained on some huge (millions of samples) dataset is “specialized” on different data and this further training requires much fewer resources.

In our case, we started from a ResNet101 (a convolutional neural network that is 101 layers deep) pre-trained on the MS COCO (Microsoft Common Objects in Context), a dataset containing hundreds of thousands of images belonging to 80 different classes [50]. Basically, a network trained to be able to detect objects belonging to the 80 different classes of the MS COCO has been retrained to specialize on the grape class. The availability of pre-trained weights for MS COCO make easier to start the training since those weights can be used as a starting point to train a variation on the network. We used Google Colab, a cloud service that provides free access to computing resources including GPUs. The experiments were executed by a virtual machine with 12 GB of RAM, an Nvidia graphic card (Tesla P100-16GB), and 68 GB of disk space. We performed fine-tuning (Goodfellow et al. [21]) using the GrapeCS-ML dataset images. The dataset was divided into a train (set 1, containing more than 1000 images), validation (set 2, containing more than 500 images), and test (sets 3, 4, and 5, containing nearly 500 images); see Table 2 for further details.

The internal dataset collected at the University of Sassari contains 451 images from all around Sardinia. The photos collect images of clusters of the main cultivars grown on the island. Specifically, of the 451 photos, almost 200 are *Cannonau* and *Vermentino* cultivars. Every single photo represents a different biotype or clone obtained following two important experimental works on mass and clonal selection for cv. *Cannonau* [51] and varietal comparison for cv. *Vermentino* [52].

The other photos were collected mainly in collection vineyards of the University of Sassari where all the regional varieties registered in the *national register of Italian vine varieties* are grown [53].

Regarding the presence of different varieties, we point out the main difference with respect to similar works. The introduction of several different varieties will probably contribute to the generalization, but it is difficult to evaluate this contribution if examples of all the varieties are present in train, validation and test at the same time. In our work we have done something totally different since there are notable differences, in terms of varieties, between train, validation, and test (in the Australian GrapeCS-ML dataset), and above all, a second test dataset was created with further different varieties (the Italian internal dataset).

Images dimensions in the first four sets of the GrapeCS-ML dataset are almost always 480 × 640 or 640 × 480. Conversely, images dimensions in the set 5 of GrapeCS-ML dataset and in the internal dataset vary a lot, from 480 × 640 up to 3024 × 4032 or 4608 × 3456 and many more (see Table 3). Since those sets are both used to test the system, consistent results could prove the robustness even towards considerable variations in size. In order to be processed by the Mask R-CNN framework all the images are automatically resized to 1024 × 1024 pixels. The aspect ratio is preserved, so if an image is not square it is padded with zeros.

To expand the size of the training part of the dataset, we used a technique called “data augmentation” through which many modified versions of the images in the dataset are created by horizontally flipping, translating, and adding artificial blur and contrast (a few augmentation examples are shown in Figure 6).

This technique allows to considerably extend the number of samples presented to the network during the training phase and, accordingly, to increase its detection and generalization capabilities. Moreover, variations in blurring, color, and brightness are a major problem in the field of computer vision. While in other works the authors try to limit those variations as much as possible, on the contrary, we have tried to include as many variations as possible in our training using dataset augmentation, so that the system “learns” to detect a grape bunch under as many as possible different conditions. It is worth noting that we only used set 1 for train due to the highest numerosity; more than 1000 images which is half of the entire GrapeCS-ML Dataset. The training of the network with a single variety, which could quickly lead to overtraining, is balanced by the use of data augmentation and the high number of varieties present in the validation set.

## 3. Results

### 3.1. Performance Evaluation

To evaluate the effectiveness of the proposed approach for bunches detection, we used the Intersection over Union (IoU) measure (Equation (1)), which allows us to estimate the precision in the overlap between a bounding box obtained by the classifier and that defined as ground truth that is the one hand drawn during the ‘labelling’ process.
(1)$$\mathrm{IoU}=\frac{\mathrm{Ground}\;\mathrm{Truth}\cap^{\;}\mathrm{Prediction}}{\mathrm{Ground}\;\mathrm{Truth}\cup^{\;}\mathrm{Prediction}}$$

This measure is given by the ratio between the intersection and the union of the surfaces of the two bounding boxes (Figure 7), and it is positively evaluated if it exceeds a given threshold value (usually 0.5, but other values can also be considered [22]). In Figure 8, two examples of IoU are presented, one higher and the other lower than 0.5.

The following values are defined:TP (True Positive): bounding boxes correctly detected (IoU > 50%);FP (False Positive): bounding boxes wrongly detected (there are no bunches or IoU < 50%);FN (False Negative): bounding boxes not detected where the bunches are present.

Precision (the ratio between the number of correctly detected bunches and the total number of objects detected as bunches in the image) and Recall (the ratio between the number of correctly detected bunches and the number of all the bunches present in the image) can therefore be calculated for each class as
(2)$$\mathrm{precision}=\frac{\mathrm{TP}}{\mathrm{TP}+\mathrm{FP}}$$
(3)$$\mathrm{recall}=\frac{\mathrm{TP}}{\mathrm{TP}+\mathrm{FN}}$$

Since each bounding box is detected with a certain probability, values of this probability higher than a certain threshold represent the more probable grape’s locations. As this threshold grows from 0.0 to 1.0, all possible Precision and Recall values are obtained. These values can be used to plot, for each image, a curve with Precision as *y* value and Recall as *x* value called the Precision–Recall Curve. The most important points are those for which there is a value change for Precision or for Recall. The purpose of this procedure is the computation of the area below this curve that is called Average Precision (AP) and can be used as a measure of the detection performance on the image. In Figure 9, an example of the Precision–Recall curve obtained during our experiments is presented. The mean of all the obtained values is known as mean Average Precision (mAP) and is among the most used metrics in the field of object detection.

### 3.2. Loss Function

An important step in the training of a model is the selection of a loss function to evaluate the network performances. Among many possible values, we chose the sum of losses obtained from the three different outputs of the Mask R-CNN framework as they represent the best compromise between the three different losses:*L* = *L_cls_* + *L_box_* + *L_mask_*(4)

$L_{cls}$ is the classification loss, $L_{box}$ is the bounding-box loss, and $L_{mask}$ is the mask loss as described in [28].

It is well known that during the training process, the validation loss is essential in choosing when to stop. As a matter of fact, if the training loss (evaluated on the train dataset) indicates how well the system is learning to perform the object detection on the training set (that is the already known data), the validation loss (evaluated on the validation dataset) explains how much the system is able to generalize the detection capability on never seen data. Figure 10 shows the training and validation loss values obtained by our system. The number of epochs, which is the number of times the learning algorithm update the model by analyzing the entire training dataset, is used as a temporal scale.

### 3.3. Detection Results

The result obtained by applying the detector on the test samples was a mAP value of 92.78%. It means that a large majority of the bounding boxes have been correctly detected. In their works, Reis et al. [34] correctly identified 91% of white grapes and 97% of red grapes, Diago et al. [35] obtained a global Recall of 74.3% and a global Precision of 92.9% for flower detection in grapevine inflorescence, Liu and Whitty [43] detected bunches with an average accuracy of 88.0% and a recall of 91.6%, Aquino et al. [40] detected flowers with an average Precision of 0.8338 and a Recall of 0.8501, and Liu et al.’s [36] average detection performance was an Accuracy of 0.8683 and an F1 Score of 0.9004. Unfortunately, it is difficult to make a direct comparison among all those results and ours. Indeed, the evaluated metrics are not always the same and, most importantly, the experimental set-up used to retrieve the images is different (usage of different cameras and acquisition with different light conditions) as well as the vines varieties are different. Despite this, it can be observed that the values we obtained are competitive with most of the works presented since, despite the different metrics, the recognition rate almost never exceeds 92%. The only case where the recognition rate seems higher is when images are captured at night using the camera’s internal flash with very little light/brightness variation [34]. A more reliable comparison can be made with the results obtained by Seng et al. [47] on the GrapeCS-ML Dataset although it is not clear which images were used for training, for validation, and for testing. By applying six different algorithms on four different color spaces, the highest classification rate they were able to achieve was 84.4% for white cultivars and 89.1% for red cultivars. In our work, there is no distinction between white and red cultivars, but with a mAP value of 92.78%, we can claim that our results are competitive with what is currently the state of the art.

The detailed results are shown in the second column of Table 4 (train complete, with augmentation). The validation and test values are very similar, proving the generalization capability of the system. Since the test dataset is composed of three subsets of the GrapeCS-ML Dataset, we also present the mAP of each of them. The considerable variation in the results is because the images in the three sets have very different characteristics. As shown in Figure 11, while in set 3 the bunches images are usually well defined and easy to detect, in set 4 and, even more, in set 5 there is a greater overlap between different bunches. Furthermore, the prevalence of red grapes in set 3 makes the detection much easier compared to the detection of white grapes, more similar in color to the surrounding vegetation.

To assess the generalization capability of the proposed framework, we also tested the system on our internal dataset. Since, concerning the GrapeCS-ML dataset, our images contain different grape varieties, different vegetation, and different colors, it would be important to replicate on our dataset results similar to those obtained with the original test. As shown in Table 4, we obtained an 89.90% mAP that it is only slightly smaller than the other values.

In order to determine the importance of the size of the dataset used for train and to determine the importance of the augmentation techniques, we performed two more training. We followed the described workflow, but we used only a reduced set of the original train (10% of the training images randomly selected) in one case with and in the other without the dataset augmentation. As it could have been expected, in the third and fourth column of Table 4 we observe a decrease in the mAP values especially in the experiments performed without augmentation. The obtained results prove the importance of a high number of images in the train but also of the use of augmentation techniques. Most of the results show a decrease between 3% and 5%, passing from the value obtained with the complete train, to those with the reduced train, to those with the reduced train and without augmentation. Exceptions are the always very high values obtained for set 3 for which the decrease is limited to values around 1% and those on set 5 and the internal dataset which are much lower than the others. It is particularly important to highlight the different values obtained for the internal dataset: if the considerable reduction in the number of train images causes a limited reduction in performance (around 4%), the absence of augmentation leads to a drop in performance (more than 15%).

Since the overlap between bunches and the presence of smaller bunches is probably the factor that reduced the detection capability, we also divided the different sets into subsets based on the number of ground truth objects in each image. The results presented in Table 5 as expected prove that the detection capability usually decreases as the number of objects in the image increases. It is worth noting that by analyzing the results from this different perspective, those that seemed evident differences between the various datasets are considerably reduced. Whatever the dataset, when there is just a single bunch in the image, the detection rate is always high.

## 4. Discussion

As stated before, the system is not error-free, since some bounding boxes are not detected at all, and others are not correctly detected, meaning that their IoU, with the ground truth, is lower than 0.5. In Figure 12, an example is shown of correct detection on a test image (the green boxes represent the ground truth, while the blue ones are the detection results) since the IoU is clearly greater than 0.5. Other examples, in Figure 13, show some of the typical problems of object detection. In Figure 13a, only one out of two bounding boxes is correctly detected. In Figure 13b,c, the two bunches are detected but as a single element. This is one of the cases in which the error can be considered as “less severe”, since the area containing the bunches has been correctly detected. Unfortunately, when many bunches stay so close together inside the same image, they are difficult to distinguish. In Figure 13d, the picture is out of focus and only the larger of the two bunches has been correctly detected. These examples confirm the results presented in Table 5 since a single bunch is almost always correctly detected. Most of the errors are due to the presence of bunches that are too small and out of focus or to the inability to distinguish partially overlapping bunches.

Those obtained with the internal dataset can be considered excellent results due to the considerable difference between the grape varieties images in this dataset, and those used to train the system. In Section 2, we stated that our aim was the development of a grape detector able to analyze images automatically acquired in a generic vineyard. We could even claim that those are the most important results presented in this work.

As in the previous cases, most of the errors are due to the incorrect detection of overlapping bunches (Figure 14a), others are caused by the inability to correctly detect shaded parts (upper-left box in Figure 14b). Another error is the incorrect detection of some leaves as bunches (Figure 14c) and it is probably due to the difference between the Sardinian grape varieties and those in the training dataset. In this case, the difference between the leaves in the image and all of those previously shown, belonging to the GrapeCS-ML dataset, is evident. This type of error can be significantly reduced by training and testing the system with images collected in the same geographic area or, even better, in the same vineyard, but the focus of this work was, on the contrary, the analysis of a generalized capability of the framework. This capability is shown, for example, in the two images of the internal dataset in Figure 15, where the same bunches are depicted. It is worth noting that, despite the image in Figure 15a is considerably overexposed, all the clusters have been correctly recognized as in Figure 15b. The ability to correctly detect grape bunches of varieties never seen before under uncontrolled lighting conditions is the main novelty of this work. Once again we emphasize the fact that, even in the most difficult cases, single bunches are almost always correctly detected as shown in Table 5.

Details of yield estimations involving traditional methods such as the lag phase method and others can be found in [54,55]. The quality assessment by visual inspection scales poorly to large vineyards and appears inaccurate due to the subjectivity of the human evaluator [56]. Moreover, these methods are expensive, inaccurate (if the yield is unevenly distributed across the vineyard), laborious, and time-consuming since they require a manual measurement of specific cluster features [57]. The precise knowledge of the number of bunches and their position in the vineyard would automate such activities.

## 5. Conclusions

In this paper, a detector of grape bunches based on the Mask R-CNN framework was presented. The GrapeCS-ML dataset was used to train the system and to evaluate its performances. We also tested the detector on an internal dataset collected in several Sardinian vineyards during the last decade since our main goal was the training of a system capable of detecting bunches regardless of both the grape variety and its geographical location.

The presented results are promising since most of the bunches were correctly detected and many of the errors were only due to the incorrect detection of two adjacent bunches as one. In fact single bunches were usually correctly detected even in the most difficult cases, despite problems as shadowing and overexposition. It is worth noting that the importance of the presented methodology is that good results are obtained not only on the GrapeCS-ML database, which has been used to train the system, but also on our internal dataset, confirming the portability of the proposed methodology to different scenarios. This is something novel at the state of the art, where methodologies are normally customized over a precise context of application and not proved to be portable.

This approach is suitable to be employed, after an appropriate training, in multiple scenarios of fruit detection and tracking from autonomous systems, reducing the subjectivity of the human evaluator during the visual quality assessment and optimize monitoring operating times. Indeed, the achieved results represent a valuable first results within the activities of the Comp4Drones (C4D) project. Starting from images’ collection, it would be possible to more precisely monitor the development of the grapes, detect the diseases, estimate the yield in terms of quantity and quality, and predict the appropriate time for harvesting. In C4D the idea is to allow running those types of analysis both off-line and on-line. In this regard, as future work, we have planned a new set of training with different frameworks, starting from the YOLO detector, which is known to be less precise but much faster in order of images analyzed per second with respect to the Mask R-CNN. Such a characteristic makes YOLO particularly suitable for the usage on embedded platforms, such as co-processing units acting as companion computers [58,59], which will allow advanced on-line processing on-boards of Unmanned Autonomous/Ground Vehicles. Additionally, as future work, the teamwork plans also to collect a new set of images in the vineyards where field trials will be carried out in the coming years.

## Figures and Tables

**Figure 1 sensors-21-03908-f001:**
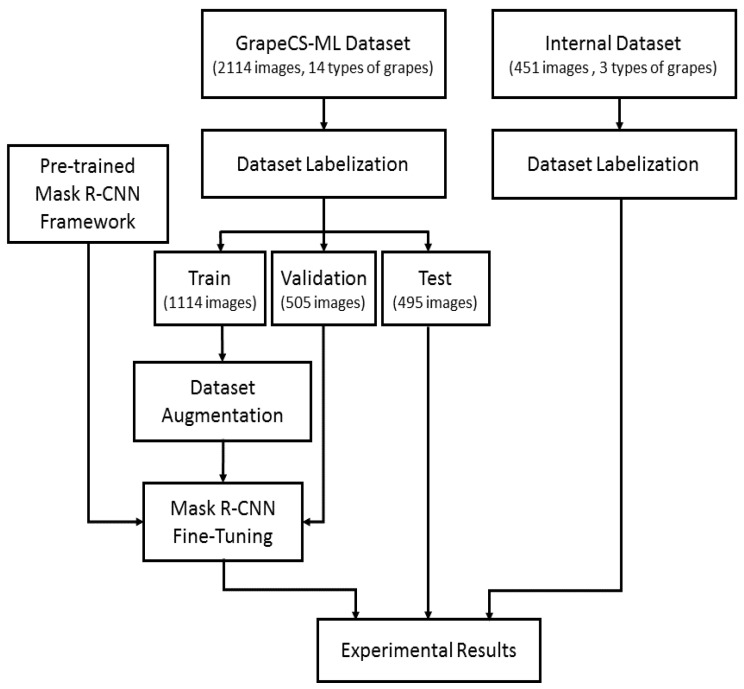
Detailed workflow of the proposed methodology. After the labeling, the data set is divided in train, validation, and test. A pre-trained Mask R-CNN framework is fine-tuned using the augmented train set and the validation set. The experimental results are obtained by applying the detector to both the test set and our internal dataset.

**Figure 2 sensors-21-03908-f002:**
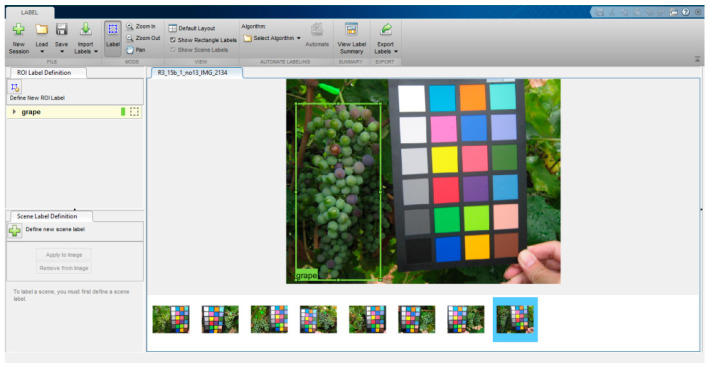
MATLAB Image Labeler used in the labeling process. For each image the smallest bounding box was hand drawn around every bunch of grapes.

**Figure 3 sensors-21-03908-f003:**
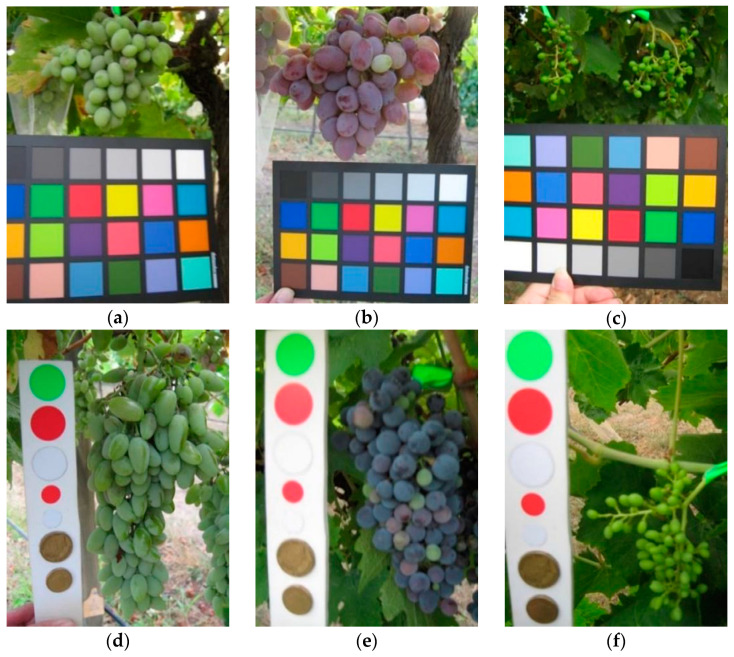
Samples images from GrapeCS-ML dataset 2: (**a**–**c**) include a color reference; (**d**–**f**) contain a volume reference.

**Figure 4 sensors-21-03908-f004:**
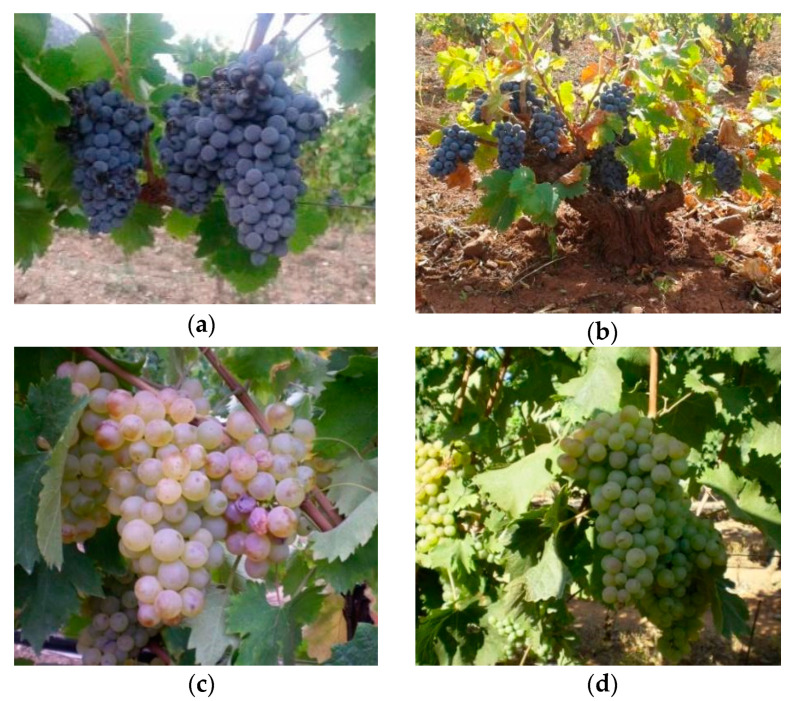
Samples images from our internal dataset: (**a**) cv. Cannonau; (**b**) cv. Cagnulari; (**c**,**d**) cv. Vermentino with different stage of maturation.

**Figure 5 sensors-21-03908-f005:**
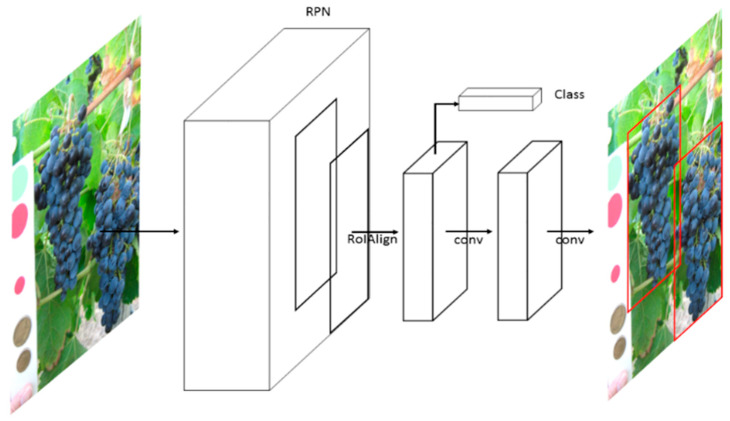
MaskR-CNN framework (He et al. [28]). In this two-stage procedure, the first stage, called Region Proposal Network (RPN), estimates the position of bounding boxes. The second stage performs a classification, a bounding box regression, and extracts a binary mask.

**Figure 6 sensors-21-03908-f006:**
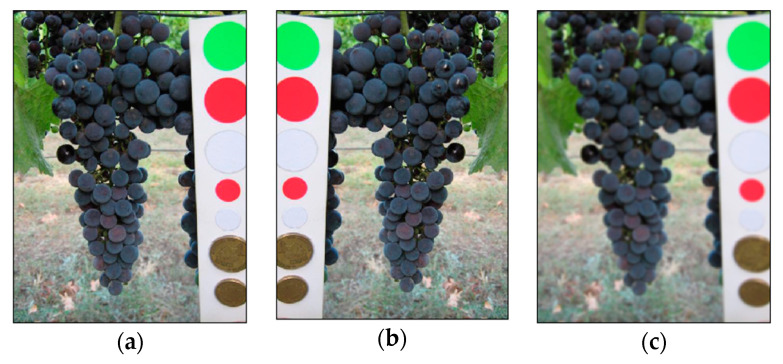
Examples of train dataset augmentation: (**a**) original image; (**b**) horizontal flipping; (**c**) image blurring.

**Figure 7 sensors-21-03908-f007:**
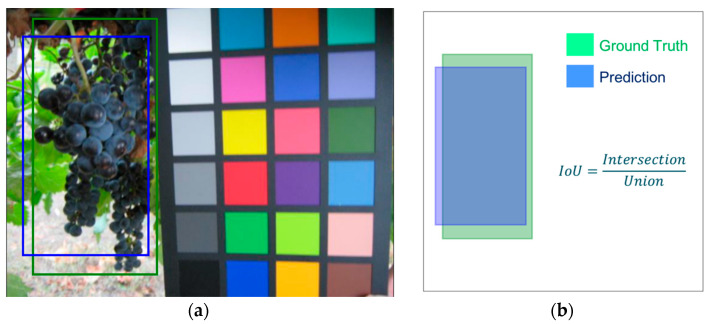
Evaluation of the IoU—Intersection over Union. This value is the ratio between the intersection and the union of the surfaces of the blue bounding box obtained by the classifier (Prediction) and the green one hand drawn during the ‘labelling’ process (Ground Truth). In (**a**) a sample image, in (**b**) a description of the calculation process.

**Figure 8 sensors-21-03908-f008:**
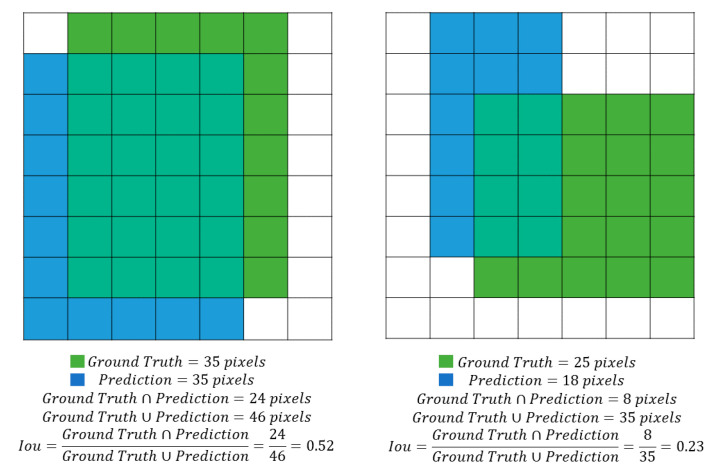
Two examples of IoU. In the example on the left the ratio between intersection and union of the ground truth and prediction bounding boxes is higher than 0.5 (0.52) while in the example on the right the ratio is lower (0.23).

**Figure 9 sensors-21-03908-f009:**
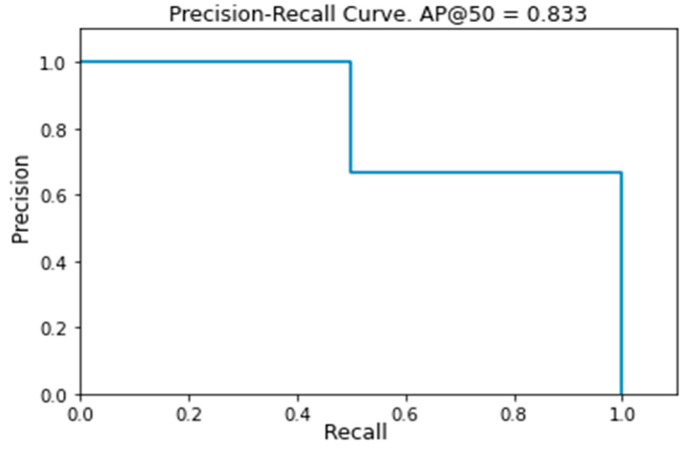
Example of Precision–Recall curve obtained during our experiments. The Average Precision, that is the area below the curve, has a value of 0.833. In this example there are three Precision or Recall value changes, but that number of changes could be different for each image.

**Figure 10 sensors-21-03908-f010:**
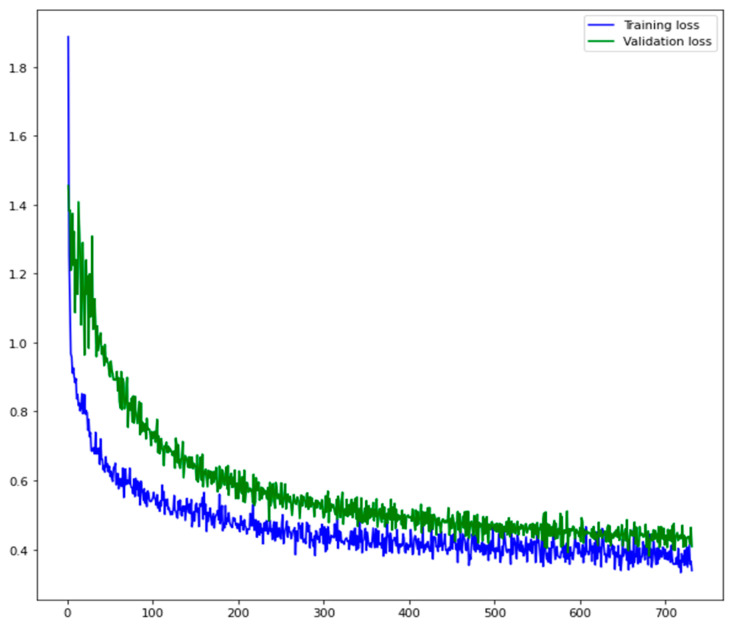
Training and validation loss profile over the number of epochs, which is the number of times the learning algorithm update the model by analyzing the entire training dataset. The two curves show the performance improvement on training and validation data.

**Figure 11 sensors-21-03908-f011:**
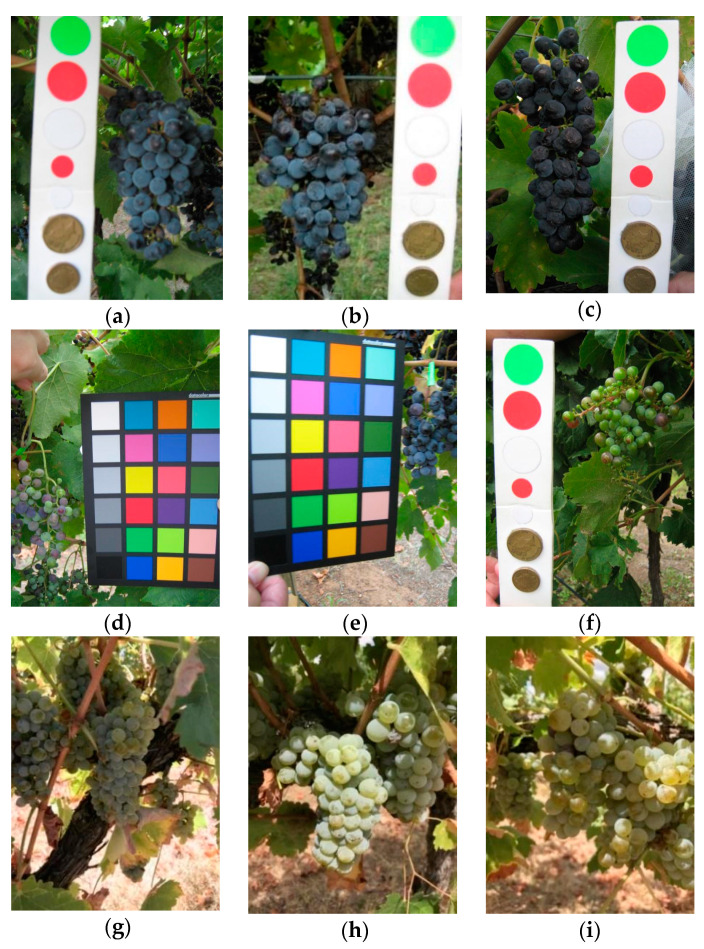
Images from the three GrapeCS-ML subsets included in the test: (**a**–**c**) set 3; (**d**–**f**) set 4; (**g**–**i**) set 5.

**Figure 12 sensors-21-03908-f012:**
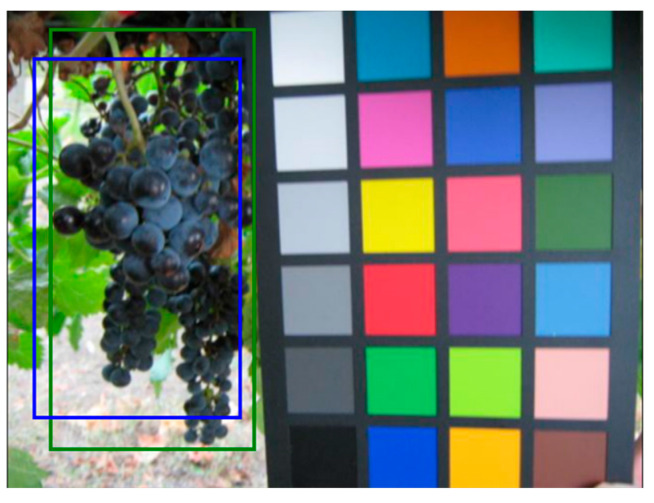
Example of correct detection on a test image from the GrapeCS-ML dataset. The green box represents the ground truth while the blue one is the detection results. The IoU of the two boxes is greater than 0.5.

**Figure 13 sensors-21-03908-f013:**
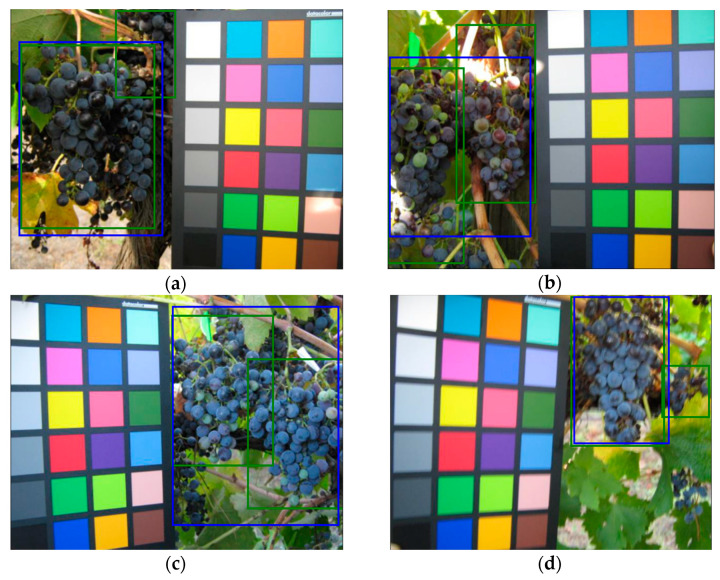
Examples of errors in the GrapeCS-ML dataset. The green boxes represent the ground truth while the blues ones are the detection results. In (**a**) only one out of two bounding boxes is correctly detected, in (**b**,**c**) the two bunches are detected but as a single element, in (**d**) only the larger of the two bunches is correctly detected.

**Figure 14 sensors-21-03908-f014:**
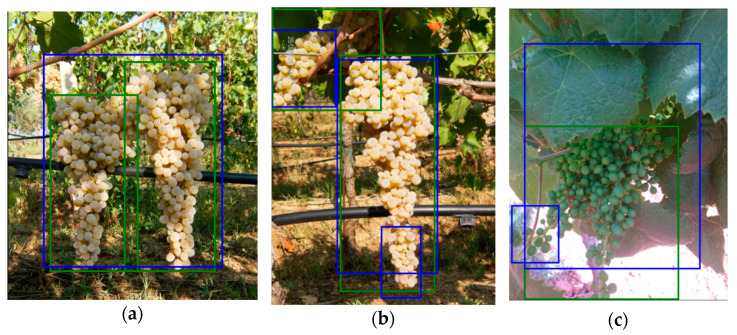
Examples of errors in the internal dataset. The green boxes represent the ground truth while the blues ones are the detection results. In (**a**) the incorrect detection of overlapping bunches, in (**b**) undetected shaded parts, and in (**c**) leaves incorrectly detected as bunches.

**Figure 15 sensors-21-03908-f015:**
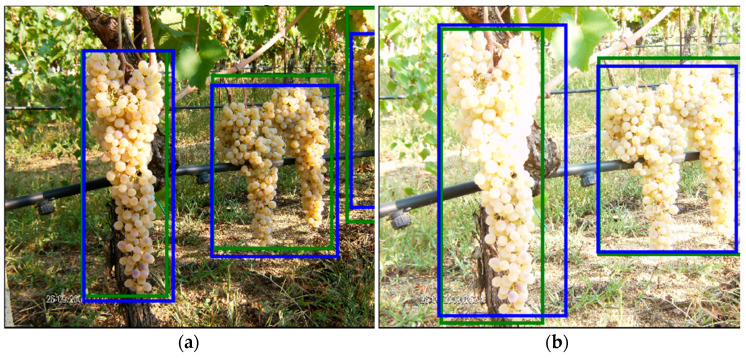
Example of same bunches correctly detected in two similar images. The image (**a**) is significantly overexposed compared to the image (**b**). The green boxes represent the ground truth while the blues ones are the detection results.

**Table 1 sensors-21-03908-t001:** Comparison between the main characteristics of grape detection proposed methods.

Reference	Fully Automated Detection Process	Large Data Set (More Than a Thousand)	Large Grape Variety (More Than Ten)
[34]	Yes (by camera internal flash at night)	No (190 images of white grapes, 35 images of red grapes)	No (Port)
[35]	No (using a uniform background of black color)	No (90 images)	No (Tempranillo, Graciano, and Carignan)
[36]	Yes	Yes (thousands of images extracted from hundreds of videos)	No (Chardonnay and Shiraz)
[38]	Yes	No (70 images)	No (Tempranillo)
[39]	Yes (with artificial illumination at night)	No (40 images)	No (Flame Seedless)
[40]	No (capturing inflorescences facing the Sun and casting a shadow on the scene to create a homogeneous illumination)	No (40 images)	No (Airen, Albariño, Tempranillo, and Verdejo)
[41]	Yes	No (139 images)	No (Riesling, Pinot Blanc, Pinot Noir, and Dornfelder)
[42]	No (using a dark cardboard behind the cluster)	No (152 images)	Yes (Tempranillo, Semillon, Merlot, Grenache, Cabernet Sauvignon, Chenin Blanc, and Sauvignon Blanc)
[43]	Yes	No (160 images)	No (Shiraz and Cabernet Sauvignon)
[44]	Yes (with natural illumination, flash illumination, and cross-polarized flash illumination)	Yes (more than one thousand images)	No (Traminette, Riesling, Chardonnay, Petite Syrah, Pinot Noir, and Flame Seedless)
[45]	Yes	Yes (more or less 100,000 images)	No (Petite Syrah and Cabernet Sauvignon)
[47]	Yes	Yes (GrapeCS-ML dataset: more than 2000 images)	Yes (Merlot, Cabernet Sauvignon, Saint Macaire, Flame Seedless, Viognier, Ruby Seedless, Riesling, Muscat Hamburg, Purple Cornichon, Sultana, Sauvignon Blanc, Chardonnay, Shiraz, Pinot Noir)
This work	Yes	Yes (GrapeCS-ML dataset: more than 2000 images + Internal dataset: 451 images)	Yes (Merlot, Cabernet Sauvignon, Saint Macaire, Flame Seedless, Viognier, Ruby Seedless, Riesling, Muscat Hamburg, Purple Cornichon, Sultana, Sauvignon Blanc, Chardonnay, Shiraz, Pinot Noir, Vermentino, Cannonau (i.e., Granache), Cagnulari (i.e., Graciano))

**Table 2 sensors-21-03908-t002:** Number of images contained in the GrapeCS-ML Dataset and in the internal dataset.

GrapeCS-ML Dataset		
Train	Set 1	1114 images
Validation	Set 2	505 images
Test	Set 3	204 images
	Set 4	242 images
	Set 5	49 images
Internal Dataset		451 images

**Table 3 sensors-21-03908-t003:** Numerosity (in brackets) per different size of the images contained in the GrapeCS-ML dataset and in the internal dataset.

GrapeCS-ML Dataset	
Set 1	480 × 640 (1102), 640 × 480 (7), 1200 × 1600 (5)
Set 2	480 × 640 (253), 640 × 480 (198), 1200 × 1600 (28), 1600 × 1200 (26)
Set 3	480 × 640 (81), 640 × 480 (81), 1200 × 1600 (21), 1600 × 1200 (21)
Set 4	480 × 640 (35), 640 × 480 (206)
Set 5	640 × 480 (1), 3024 × 3024 (12), 3024 × 4032 (36), 3402 × 3752 (1)
Internal Dataset	360 × 640 (1), 480 × 640 (29), 640 × 480 (17), 1600 × 2128 (2), 1904 × 2528 (3), 2048 × 1536 (36), 2112 × 2816 (23), 2304 × 3072 (1), 2320 × 3088 (120), 2560 × 1536 (3), 2816 × 2112 (139), 3072 × 2304 (9), 3088 × 2320 (43), 3456 × 4608 (2), 4160 × 2340 (1), 4608 × 3456 (22)

**Table 4 sensors-21-03908-t004:** Experimental results on both GrapeCS-ML and our internal dataset. The detector has been trained in three different ways: using the entire set 1 as train, with dataset augmentation; using only 10% of set 1 as a train, with dataset augmentation; using only 10% of set 1 as a train, without dataset augmentation.

	mAP		
Dataset Name	Train Complete, with Augmentation	Train 10%, with Augmentation	Train 10%, without Augmentation
Validation (Set 2)	93.97%	90.95%	85.24%
Test (Set 3 + Set 4 + Set 5)	92.78%	90.98%	87.65%
Set 3	98.77%	98.69%	97.30%
Set 4	89.18%	86.70%	83.40%
Set 5	85.64%	80.07%	68.44%
Internal Dataset	89.90%	86.41%	70.75%

**Table 5 sensors-21-03908-t005:** Experimental results on both GrapeCS-ML and our internal dataset based on the number of bunches present in the images. After each mAP value, in brackets, the number of examined images is shown.

Dataset Name	mAP (Total Number of Images)					
	1 bunch	2 bunches	3 bunches	4 bunches	5 bunches	6 bunches
Validation (Set 2)	98.85% (369)	82.61% (126)	57.22% (10)			
Test (Set 3, 4, 5)	99.75% (395)	65.41% (73)	72.59% (15)	51.72% (8)	60.00% (2)	64.63% (2)
Set 3	100.00% (195)	72.22% (9)				
Set 4	99.45% (181)	57.70% (53)	65.28% (8)			
Set 5	100.00% (19)	96.97% (11)	80.95% (7)	51.72% (8)	60.00% (2)	64.63% (2)
Internal Dataset	96.79% (218)	85.39% (166)	76.11% (46)	80.89% (17)	99.17% (4)	

## Data Availability

The GrapeCS-ML database is openly available at doi:10.26189/5da7a8603c55c. The images of the Internal Dataset were collected by different teams and were granted to us for internal use only.

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
