# Peer review of "In-Field Automatic Detection of Grape Bunches under a Totally Uncontrolled Environment"

_sensors, 2021, doi:10.3390/s21113908_

Round 1
Reviewer 1 Report
In this paper a detector of grape bunches based on a deep convolutional neural network is presented.
The paper includes and adequate stat of the art, identifying the main limitations of previous proposals and summarizing the proposed methods in a single Table considering the process automation, the size of the dataset and the grape variety.
With respect to the Material and Methods section the dataset was adequately labeled and divided in training, validation and test subsets considering the GrapeCS-ML Dataset together with an additional internal dataset. However, it is snot clear why it was only used for the training the set 1 (which only includes Merlot type) and for the validation the set 2 (chich includes several types). Please clarify.
The main issue with the paper is the comparison with the state of the art. As stated, the state of the art is well presented qualitatively, but there is a lack of a quantitative comparison with previous algorithms in terms of precision. This aspect must be included in order to show how the paper contributes to the state of the art.
Author Response
In this paper a detector of grape bunches based on a deep convolutional neural network is presented.
The paper includes and adequate stat of the art, identifying the main limitations of previous proposals and summarizing the proposed methods in a single Table considering the process automation, the size of the dataset and the grape variety.
Question
With respect to the Material and Methods section the dataset was adequately labeled and divided in training, validation and test subsets considering the GrapeCS-ML Dataset together with an additional internal dataset. However, it is snot clear why it was only used for the training the set 1 (which only includes Merlot type) and for the validation the set 2 (chich includes several types). Please clarify.
Answer:
We only used set 1 for train due to the highest numerosity, more than 1000 images that is half of the entire GrapeCS-ML Dataset. The training of the network with a single variety, which could quickly lead to overtraining, is balanced by the use of data augmentation and the high number of varieties present in the validation set.
We added these remarks in lines 314-318.
Question
The main issue with the paper is the comparison with the state of the art. As stated, the state of the art is well presented qualitatively, but there is a lack of a quantitative comparison with previous algorithms in terms of precision. This aspect must be included in order to show how the paper contributes to the state of the art.
Answer:
We presented some of the state of the art results in lines 383-390 but in lines 390-394 we also stated:
“Unfortunately, it is difficult to make a direct comparison among all those results and ours. Indeed, the evaluated metrics are not always the same and, most importantly, the experimental set-up used to retrieve the images is different (usage of different cameras and acquisition with different light conditions) as well as the vines varieties are different.”
A shared dataset as the GrapeCS-ML could allow a more reliable comparison, but only if the training process and the used algorithms are fully described. The paper [47] in the references present the dataset and also include several experiental results (see tables 7 and 8 in [47]) but, unfortunately, it is not explained which images are used for training, for validation and for testing.
Reviewer 2 Report
This paper presents the method of automatic detection of grape bunches capable of applying under uncontrolled environments. The introduction is giving very sufficient background and references well. It is recommended to be accepted for the publication with small revisions.
How about "grape bunches" instead of "bunches" in the title?
Please provide the full name of NIR in the line number of 65.
Is it a typo, "ReliefF" in the line number of 134?
It could be more useful if the reason of fine-tuning effect for the student reader is presented around the line number of 257.
Moreover, some details of MS COCO's pre-trained model are required to satisfy the curiosity such as kinds of labels and so on.
The input method of different size of images has to be explained.
Table 5 needs more analysis and conclusions.
Author Response
This paper presents the method of automatic detection of grape bunches capable of applying under uncontrolled environments. The introduction is giving very sufficient background and references well. It is recommended to be accepted for the publication with small revisions.
Question
How about "grape bunches" instead of "bunches" in the title?
Answer:
We agree with the reviewer and the title has been modified as suggested.
Question
Please provide the full name of NIR in the line number of 65.
Answer:
We included the full name and we thanks the reviewer for the suggestion.
Question
Is it a typo, "ReliefF" in the line number of 134?
Answer:
No, it is not a typo. The ReliefF is the name of the algorithm adopted for feature selection as described in the cited work (the paper [43] in the references).
Question
It could be more useful if the reason of fine-tuning effect for the student reader is presented around the line number of 257.
Moreover, some details of MS COCO's pre-trained model are required to satisfy the curiosity such as kinds of labels and so on.
Answer:
In lines 264-267 we added:
“Basically, a network trained to be able to detect objects belonging to the 80 different classes of the MS COCO has been retrained to specialize on the grape class. The availability of pre-trained weights for MS COCO make easier to start the training since those weights can be used as a starting point to train a variation on the network.”
Question
The input method of different size of images has to be explained.
Answer:
In lines 297-300 we added:
“In order to be processed by the Mask R-CNN framework all the images are automatically resized to 1024x1024 pixels. The aspect ratio is preserved, so if an image is not square it is padded with zeros.”
Question
Table 5 needs more analysis and conclusions.
Answer:
We added several remarks in lines 456-459, 487-488 and 511-512.
Reviewer 3 Report
The authors use deep learning and computer vision to detect the fruits. The background, method, and results are appropriate. The only concern is about the claim: fruit detection. The instance discussed in the paper only one fruit: grape. And this fruit is difficult to generalize to other fruits based on the theories and methods mentioned in the paper. So, I suggest modifying the abstract and the claims. Then, can claim that the possibility of the methods used in other fruit detection.
Author Response
The authors use deep learning and computer vision to detect the fruits. The background, method, and results are appropriate. The only concern is about the claim: fruit detection. The instance discussed in the paper only one fruit: grape. And this fruit is difficult to generalize to other fruits based on the theories and methods mentioned in the paper. So, I suggest modifying the abstract and the claims. Then, can claim that the possibility of the methods used in other fruit detection.
Answer:
We thanks the reviewer and we modified abstract and claims as suggested (lines 18 and 518).
Round 2
Reviewer 1 Report
Thanks for the reviewed version. Please note that even if it is difficult to compare the proposal with respect to the state of the art it is required to judge the contribution.
Author Response
Comments and Suggestions for Authors
Thanks for the reviewed version. Please note that even if it is difficult to compare the proposal with respect to the state of the art it is required to judge the contribution.
Answer:
We added a comparison with the other presented works and a more reliable comparison with the paper [47] but we strongly doubt its validity. In fact, in the paper [47], it is not described the procedure followed by the authors to divide the data into train, validation and test and it is not even specified how the Classification rate is calculated!
In lines 387- 398 we added:
Despite this, it can be observed that the values we obtained are competitive with most of the works presented since, despite the different metrics, the recognition rate almost never exceeds 92%. The only case where the recognition rate seems higher is when images are captured at night using the camera's internal flash with very little light/brightness variation [34]. A more reliable comparison can be made with the results obtained by Seng et al. [47] on the GrapeCS-ML Dataset although it is not clear which images were used for training, for validation and for testing. By applying 6 different algorithms on 4 different color spaces, the highest classification rate they were able to achieve was 84.4% for white cultivars and 89.1% for red cultivars. In our work there is no distinction between white and red cultivars but, with a mAP value of 92.78%, we can claim that our results are competitive with what is currently the state of the art.
This manuscript is a resubmission of an earlier submission. The following is a list of the peer review reports and author responses from that submission.
Round 1
Reviewer 1 Report
The paper is interesting, although few sections require some additions.
You summarize the score for your method "91% average detection rate". Such metric is not defined in the paper, could you precise which metric was used to calculate this? In fact, this reported 91% is not present amongst results commented in the paper? Where it came from?
Could you clarify why haven’t you used the own "internal" dataset during the training phase and only for testing? Could this improve the generalization of the detector?
Does all images from the test dataset cover a calibration panel - is it always visible? (all of those presented in the paper do). How that fact influence the detector and the calculated scores? Do you plan the presence of a calibration panel as a required element for the final production detector?
Have you tried to train the detector on images without those calibration panels and compared the scores? Could you include comments on how this possibly influences the detection process?
You defined IoU, Prec, and Recall, but could you extend the definition of mean Average Precision (mAP) - how it could be interpreted regarding your detector?
In the beginning, the paper presents the "harder" case and says:
In the future, this tool could [...] counting efficiently fruits under the shadow, occluded by foliage, branches, or [...] some degree of overlap amongst fruits. Experimental results show a 91% average detection rate.
- Does the reported 91% average detection rate refer to this "harder case" (images taken against shadow, occluded, and overlapped fruits).
Why those images are not presented (and only those with shadows are mentioned and shown?)
Minor corrections:
The sections "3.2 Loss function" is worthless, and could be omitted
Please consider the term replacement - from "labelization" to "labeling" (example: https://firebase.google.com/docs/ml/label-images)
Could you specify how do you count True Positives in your study? Is it also calculated against IoU (like FP) using a threshold of 50%, something like IoU>50%?
Line: 323: please change: "11it" to "11 it"
Author Response
Reviewer 1
Comments and Suggestions for Authors
The paper is interesting, although few sections require some additions.
You summarize the score for your method "91% average detection rate". Such metric is not defined in the paper, could you precise which metric was used to calculate this? In fact, this reported 91% is not present amongst results commented in the paper? Where it came from?
Answer:
You’re absolutely right, that value is a mean Average Precision and it is the average between the results obtained on the GrapeCS-ML dataset (93.97%) and on the internal dataset (89.90%). We modified the sentence using “mean Average Precision” instead of “average detection rate”.
Could you clarify why haven’t you used the own "internal" dataset during the training phase and only for testing? Could this improve the generalization of the detector?
Answer:
The main reason for this choice is pushing for a completely target data agnostic methodology. We didn’t use the internal dataset during the training phase in order to test the performances of the detector on never seen images.
The insertion of those images in the training would have improved the detector, but it would not have allowed a correct evaluation of its detection capabilities on images never seen before.
Does all images from the test dataset cover a calibration panel - is it always visible? (all of those presented in the paper do). How that fact influence the detector and the calculated scores? Do you plan the presence of a calibration panel as a required element for the final production detector?
Answer:
In lines 189-191 we stated:
“A color reference or a volume reference is present in most of the images (a few examples are shown in Figure 3) but we chose to ignore this kind of information in order to obtain a fully automated detection process.”
The images in four out of five set that compose the GrapeCS-ML dataset contain a calibration panel. No calibration panel is present in the entire Internal Dataset. We have not used the information regarding panels neither in the training nor We respectfully disagree, the agriculture aspect could not (and should not) be ignored and anyway we don’t in the testing phase and we do not plan to use them in the future.
Have you tried to train the detector on images without those calibration panels and compared the scores? Could you include comments on how this possibly influences the detection process?
Answer:
The calibration panels are present in most of the GrapeCS-ML dataset images, but they’re never included in the bounding boxes and therefore they don’t influence the detection process. At the moment we have dataset campaign acquisition on-going. As soon as we will have sufficient number of images, we intend to train the detector on the new set of images without the calibration panels.
You defined IoU, Prec, and Recall, but could you extend the definition of mean Average Precision (mAP) - how it could be interpreted regarding your detector?
Answer:
As suggested, we extended the definition of mean Average Precision in lines 294-297. We also added an example of Precision-Recall curve in line 301.
In the beginning, the paper presents the "harder" case and says:
In the future, this tool could [...] counting efficiently fruits under the shadow, occluded by foliage, branches, or [...] some degree of overlap amongst fruits. Experimental results show a 91% average detection rate.
- Does the reported 91% average detection rate refer to this "harder case" (images taken against shadow, occluded, and overlapped fruits).
Why those images are not presented (and only those with shadows are mentioned and shown?)
Answer:
Harder cases will be included in future works. In fact, in the current database, just few critical examples are present. Some overlap amongst fruits is present each time two bunches are not entirely separate, as in Fig. 13. In Fig. 14.c part of the bunch is occluded by foliage.
Minor corrections:
The sections "3.2 Loss function" is worthless, and could be omitted
Answer:
We respectfully disagree: the loss function is a key part of a network training. The two curves in Fig. 10 are particularly important as they show the several steps required to reach the best possible training level.
Please consider the term replacement - from "labelization" to "labeling" (example: https://firebase.google.com/docs/ml/label-images)
Answer:
Every occurrence of the term "labelization" have been replaced with the term "labeling".
Could you specify how do you count True Positives in your study? Is it also calculated against IoU (like FP) using a threshold of 50%, something like IoU>50%?
Answer:
Yes, just like FP, the True Positives are calculated using the IoU. A bounding box is correctly detected if IoU>50%. The text in line 283 has been updated.
Line: 323: please change: "11it" to "11 it"
Answer:
We thank you for the notice, we added the missing space.
Reviewer 2 Report
Comments paper: “In field automatic detection of bunches under totally uncontrolled environment”.
Ghiani et al, 2021
Sensors 1148396
A major challenge in precision agriculture and in the field of viniculture is the correct detection of the exact number of tress, fruits, flowers, and for instance bunches of grapes. The paper describes the development a grape detector to analyse images automatically acquired by a moving vehicle. To detect vine bunches directly in the field, image analysis based on deep convolutional neural network has been exploited.
Although the paper is well written, there are quite some issues that should be attended too in order to make the paper more accessible to the audience. See the specific comments
Specific comments:
Introduction:
p.2,
- line 59-60: explain the abbreviations when they appear for the first time in the paper: R-CNN, YOLO
- line 67: CIELAB
p.3, line 122: GrapeCS-ML, explain abbreviation
Materials and methods:
p.6, line 172-173: clarify why “not all the collected images are suitable for our type of analysis” and the criteria to select the 451 images that were selected to test the trained network.
p.7, figure 5: the letters in the fig are too small and should be repositioned to make it more readable
p.8, line 206: MS COCO, explain the abbreviation and give a reference
table 2: clarify the criteria to define amount of images in the different sets; for instance are the features of the 1114 images chosen or the set 1 (train) different of these of the 505 images of set 2 (validation)?
Results:
p.9, line239 and fig 8, p.10: “(usually 0.5, but other values can also be considered)”: on what is this threshold based and what are the arguments to use other values?
Fig.8: I suppose these are “theoretical examples”, how do you proceed in practice (on real samples) to choose in one case 35/35 pixels ad in another case 25/18 pixels? You explain it more or less in the text following the figure (line 248 to 263), please present an example of the so called “Precision-Recall curve”.
p.11, fig 9: the text in the inset in the figure is too small, make it larger.
Conclusions:
p.15, line 372: GRPACS-ML database: explain the abbreviation.
Author Response
Reviewer 2
Comments and Suggestions for Authors
Comments paper: “In field automatic detection of bunches under totally uncontrolled environment”.
Ghiani et al, 2021
Sensors 1148396
A major challenge in precision agriculture and in the field of viniculture is the correct detection of the exact number of tress, fruits, flowers, and for instance bunches of grapes. The paper describes the development a grape detector to analyse images automatically acquired by a moving vehicle. To detect vine bunches directly in the field, image analysis based on deep convolutional neural network has been exploited.
Although the paper is well written, there are quite some issues that should be attended too in order to make the paper more accessible to the audience. See the specific comments
Specific comments:
Introduction:
p.2,
- line 59-60: explain the abbreviations when they appear for the first time in the paper: R-CNN, YOLO
- line 67: CIELAB
p.3, line 122: GrapeCS-ML, explain abbreviation
Answer:
As requested, we added the meaning of all the abbreviations with the exception of GrapeCS-ML whose origin is not clear from documentation. CS could stand for Cabernet Sauvignon and ML for Machine Learning, but those are only hypotheses.
Materials and methods:
p.6, line 172-173: clarify why “not all the collected images are suitable for our type of analysis” and the criteria to select the 451 images that were selected to test the trained network.
Answer:
The available images were thousands and they were acquired all around several sardinian vineyards. Some contained the entire vineyard, others in perspective the space between two rows or an entire row imaged from one end. The purpose of our work was to train a detector able to analyze images automatically acquired by a vehicle moving between the vine rows. Therefore we only selected photos acquired between the rows at a distance of about one meter from the leaf wall. We added this information to the paper in lines 200-206.
p.7, figure 5: the letters in the fig are too small and should be repositioned to make it more readable
Answer:
We apologize for the inconvenience, the image was mistakenly resized. We restored the original size and now all the labels are much more readable.
p.8, line 206: MS COCO, explain the abbreviation and give a reference
Answer:
We explained the abbreviation (Microsoft Common Objects in Context) and added the reference to the dataset:
Lin, T.-Y., Maire, M., Belongie, S., Bourdev, L., Girshick, R., Hays, J., Perona, P., Ramanan, D., Zitnick, C.L., Dollár, P., 2015. Microsoft COCO: Common Objects in Context. arXiv:1405.0312 [cs].
table 2: clarify the criteria to define amount of images in the different sets; for instance are the features of the 1114 images chosen or the set 1 (train) different of these of the 505 images of set 2 (validation)?
Answer:
Our main goal was to train a detector using as many images as possible for the training, but without reducing too much validation and test set. The best compromise was to keep the original subset division by using the largest set (set 1) for the training and by dividing the other 4 sets in order to have similar image numbers in validation and test sets (as shown in table 2). Due to the presence of different species in the 5 subsets we can state that the features are different enough despite some overlapping (as Merlot in set 1, 2 and 4 or Cabernet Sauvignon in set 2 and 3).
Results:
p.9, line239 and fig 8, p.10: “(usually 0.5, but other values can also be considered)”: on what is this threshold based and what are the arguments to use other values?
Answer:
That particular value is among the most used in literature and in this particular case it seemed to us the most suitable for evaluating a good level of overlapping. Other widely used values are 0.75 and 0.9, but the former and especially the latter are used when a much greater precision is needed.
Fig.8: I suppose these are “theoretical examples”, how do you proceed in practice (on real samples) to choose in one case 35/35 pixels ad in another case 25/18 pixels? You explain it in the text following the figure (line 248 to 263), please present an example of the so called “Precision-Recall curve”.
Answer:
Yes, these are just theoretical examples and we could have presented different examples with different number of pixels. We added an example of Precision-Recall curve as requested (Figure 9).
p.11, fig 9: the text in the inset in the figure is too small, make it larger.
Answer:
Once again, we apologize for the inconvenience, the image was mistakenly resized. We restored the original size and now all the labels are much more readable.
Conclusions:
p.15, line 372: GRPACS-ML database: explain the abbreviation.
Answer:
It was a misspelling of the GrapeCS-ML dataset name. We restored the correct term.
Reviewer 3 Report
- The subject of the paper has put too much emphasis on the agriculture aspect of the work. As it is, the paper is not too relevant to the readers in Sensors. The paper needs to be modified in order to be suitable for Sensors reader.
- What is the exact novelty of the paper? It is not clear what new knowledge has been introduced here. As far as I know, the system is a made up of individual systems that have been proposed before and used in other applications.
- Why would a sandglass type of network can benefit such applications? There are so many variants of deep network but it is not clear at all why motivates the use of sandglass architecture. Please explain and elaborate.
- The proposed idea has been previously developed by Sulistyo. See the following work:
- “Computational deep intelligence vision sensing for nutrient content estimation in agricultural automation,” IEEE Trans. Automation Science and Engineering, 2018
- “Regularized neural networks fusion and genetic algorithm based on-field nitrogen status estimation of wheat plants,” IEEE Trans. Industrial Informatics, 2016
- The above papers also used similar non-invasive methodology of digital images and machine learning for object segmentation. How does the proposed work differ from the above work?
- Please talk about the similarity and differences of the proposed work with respect to the above papers. The literature review on non-invasive methodology is quite weak and the authors can benefit the readers with more in-depth review on non-invasive methodology.
- There is the internal dataset used in the paper? Please elaborate further on how this dataset comes about.
- Why didn’t the authors compare the proposed technique with another deep neural network?
- There are other works published. For example, the authors can look into:
- “A robust deep-learning-based detector for real-time tomato plant diseases and pests recognition,” Sensors 2017
- “Deep convolutional neural networks for mobile capture device-based crop disease classification in the wild,” Computers and Electronics in Agriculture 2018.
Author Response
Reviewer 3
Comments and Suggestions for Authors
The subject of the paper has put too much emphasis on the agriculture aspect of the work. As it is, the paper is not too relevant to the readers in Sensors. The paper needs to be modified in order to be suitable for Sensors reader.
Answer:
We understand the reviewer's observation but we think that a little more emphasis on the agronomic aspect is justified by the fact that this work have been submitted to a Special Issue of Sensors: "APRAS-AI-Empowered Self-Adaptive Federation of Platforms for Efficient Economic Collaboration in Rural Areas".
The issue is focused on the agrifood sector as stated in the “Special Issue Information”:
“...The Internet of Things (IoT) provides a unique opportunity for technology to transform many industries, including the food and agriculture sector. The agrifood sector has a rather low level of uptake of information and communications technology (ICT) and a relatively high cost of data capture...”
What is the exact novelty of the paper? It is not clear what new knowledge has been introduced here. As far as I know, the system is a made up of individual systems that have been proposed before and used in other applications.
Answer:
As we stated in the paper: “It is worth emphasizing the importance of testing the system on a dataset that contains images like those we will work on. Moreover, it would be even more important to ascertain the ability of the system to provide good detection results on images very different from those present in the training set. In fact, while in the former case we would have a well performing detector on a specific vineyard, in the latter we would have a universal detector able to work anywhere.” In the majority of the works presented the detectors were trained and tested on a limited number of vineyards and with a limited number of species. Sometimes even in the same vineyard at the same time. We know that the more train, validation and test sets are similar the better are the results. But the results will also be less useful since that detector will work as expected only in those particular vineyards, with those particular species. We trained and tested the detector with images of many different grape varieties collected in Australian vineyards and we also test it on a dataset collected in the Sardinia island (Italy). That is the main novelty of our work with respect to many similar papers.
Why would a sandglass type of network can benefit such applications? There are so many variants of deep network but it is not clear at all why motivates the use of sandglass architecture. Please explain and elaborate.
Answer:
To be honest we must admit that we do not really get the raised point. The adopted approach does not belong to the same categories of the mentioned sandglasses architecture. We simply used a Region Based Convolutional Neural Network and, in particular, the Mask R-CNN framework because of the high performances on different object detection datasets. In any case, we would be happy to provide further clarifications in future iterations if needed.
The proposed idea has been previously developed by Sulistyo. See the following work:
“Computational deep intelligence vision sensing for nutrient content estimation in agricultural automation,” IEEE Trans. Automation Science and Engineering, 2018 “Regularized neural networks fusion and genetic algorithm based on-field nitrogen status estimation of wheat plants,” IEEE Trans. Industrial Informatics, 2016
The above papers also used similar non-invasive methodology of digital images and machine learning for object segmentation. How does the proposed work differ from the above work?
Please talk about the similarity and differences of the proposed work with respect to the above papers. The literature review on non-invasive methodology is quite weak and the authors can benefit the readers with more in-depth review on non-invasive methodology.
Answer:
We respectfully disagree, it seems to us that the approach followed in those works is quite the opposite with respect to ours: their main effort was focused on the correction of the color deviations while we avoided any color analysis or preprocessing step and relied on the dataset augmentation to include in our set of images as many color differences as possible. Besides, the presented experiments were conducted in a single experimental farm, on wheat plants and they performed image segmentation instead of object detection.
In any case, we could modify the paper but you should better explain your point of view and clarify what the similarities are.
There is the internal dataset used in the paper? Please elaborate further on how this dataset comes about.
Answer:
We were able to gather thousands of images collected all around many vineyards in Sardinia (Italy). Some contained the entire vineyard, others in perspective the space between two rows or an entire row imaged from one end. The purpose of our work was to train a detector able to analyze images automatically acquired by a vehicle moving between the vine rows. Therefore we only selected photos acquired between the rows at a distance of about one meter from the leaf wall.
Why didn’t the authors compare the proposed technique with another deep neural network?
Answer:
We agree with the reviewer that this could be an interesting addition and we have already planned to do it in future, as stated in the future work discussion. Moreover, it is worth highlighting that the main focus of the work is on the performances of the data-agnostic detector.
There are other works published. For example, the authors can look into:
“A robust deep-learning-based detector for real-time tomato plant diseases and pests recognition,” Sensors 2017
“Deep convolutional neural networks for mobile capture device-based crop disease classification in the wild,” Computers and Electronics in Agriculture 2018.
Answer:
We agree and we added a citation of those (and other) works in Introduction.
Reviewer 4 Report
Abstract
It is not usual to talk in Abstract about future work (20-22). The abstract is for someone who haven't read the paper, what can be found in paper. Replace with performance and evaluation description. What about obtained results in comparison with the state-of-the art?
It is not clear what is actually proposed. Bunches detector or a grape detector (in my opinion this is not the same).
Keyword: should be reconsidered (e.g. decision support system ?)
Introduction
There is a certain review of the papers. However, these are mostly based on traditional computer vision. hereby, focus should be on methods for fruit/bunch detection using deep learning methods and these must be added. Based on this overview, the paper contribution should be stated. The author should consider whether to put detail overview in the separate section (Related work ). I suggest authors to make more detailed state-of-the art overview.
Materials and Methods
Figure 1 need better explanation in text (not in too details but the complete flow should be described); not in figure caption.
To enable further scientific research and research reproducibility, the labels must be provided in some form online (xml or json with labels for example on github). Internal dataset should be published as well.
Color reference available in the image is not completely described. What is its role? How does it affect later detector training upon such images?
Results
There are some unnecesarry information regarding IoU, precision and recall. There are standard metrics in object detection. Apart from that there is a mix of information in Results and previous section. For example loss function definition.
The comparison is simply not at appropriate level. The authors should put more effort on precise and complete comparison with other methods (and more recent ones based on deep learning).
The more detailed discussion should be carried on. For example, what are the performance of the detector on different sizes of object (in pixels)? To what extent does pretraining influence the final detector performance and so on?
Language and style
Several formatting errors can be found in manuscript (e.g. line 323)
Some of the terms are written with first capital letter, why? (precision, agriculture...)
Possible error line 68: lower than a threshold --> higher than a threshold?
Author Response
Reviewer 4
Comments and Suggestions for Authors
Abstract
It is not usual to talk in Abstract about future work (20-22). The abstract is for someone who haven't read the paper, what can be found in paper. Replace with performance and evaluation description. What about obtained results in comparison with the state-of-the art? It is not clear what is actually proposed. Bunches detector or a grape detector (in my opinion this is not the same).
Keyword: should be reconsidered (e.g. decision support system?)
Answer:
We agree, we have eliminated the reference to future work in Abstract since it is present, in a more extended form, in Conclusions.
We would have liked to propose a comparison with the state-of-the art but it was not possible due to the difference between the data analyzed in the different papers. As stated in Section 3.3, results obtained on such different data cannot be compared in any way.
We realize that the terms bunch and grape do not have the same meaning and we modified the text whose meaning could be misunderstood in that sense.
We agree with the reviewer and we removed "decision support system" from the keyword list.
Introduction
There is a certain review of the papers. However, these are mostly based on traditional computer vision. hereby, focus should be on methods for fruit/bunch detection using deep learning methods and these must be added. Based on this overview, the paper contribution should be stated. The author should consider whether to put detail overview in the separate section (Related work). I suggest authors to make more detailed state-of-the art overview.
Answer:
You are absolutely right, we have updated the state-of-the art overview by adding several papers that use deep learning methods (lines 62-87).
In the guidelines it is suggested to limit the number of sections and subsections but if it deems necessary we can add some.
Materials and Methods
Figure 1 need better explanation in text (not in too details but the complete flow should be described); not in figure caption.
To enable further scientific research and research reproducibility, the labels must be provided in some form online (xml or json with labels for example on github). Internal dataset should be published as well.
Color reference available in the image is not completely described. What is its role? How does it affect later detector training upon such images?
Answer:
Sorry, maybe we misunderstood the observation but a brief description of the flow is present in the lines 156-160. The various steps are then described in more detail on the following pages.
We totally agree on the importance of research reproducibility and we would have liked to be able to compare our results with those obtained by others on the same data. Unfortunately, the currently adopted dataset is not our property. It might not be appropriate to publish a labeling of their data at this stage and without their explicit permission. Nevertheless, we see the point and the importance of this comment; therefore, we will try to contact them in future, to check for this opportunity for future activities or extension of the presented work., Regarding the Internal Dataset the issue is the same: images were collected by different teams and were granted to us for internal use only.
In lines 189-191 we stated:
“A color reference or a volume reference is present in most of the images (a few examples are shown in Figure 3) but we chose to ignore this kind of information in order to obtain a fully automated detection process.”
The images in four out of five set that compose the GrapeCS-ML dataset contain a calibration panel. No calibration panel is present in the entire Internal Dataset. We have not used the information in the panels in any way during both training and testing phase. The calibration panels are present in most of the GrapeCS-ML dataset images but they’re never included in the bounding boxes and therefore they don’t influence the detection ]process.
Results
There are some unnecesarry information regarding IoU, precision and recall. There are standard metrics in object detection. Apart from that there is a mix of information in Results and previous section. For example loss function definition. The comparison is simply not at appropriate level. The authors should put more effort on precise and complete comparison with other methods (and more recent ones based on deep learning).
The more detailed discussion should be carried on. For example, what are the performance of the detector on different sizes of object (in pixels)? To what extent does pretraining influence the final detector performance and so on?
Answer:
We agree that the level of details needed may change depending on the reader's background. Since for the same section a reviewer asked to extend the definition of mean Average Precision and another one asked for an example of precision-recall curve, we decided to keep this level also for educational purposes.
Sorry maybe we misunderstood but we didn’t find any reference to the loss function definition in the previous section. Could you please indicate us where to act?
As already stated above, the direct comparison with other works was not possible due to the lack of a common dataset. -As future work, we plan to collect our own dataset on the field, labelize it and make it freely available as you previously suggested.
Regarding the detailed analysis of the network behavior in comparison with other networks, we agree that it could be valuable in general, but in our humble opinion not in the context of the present work. - Here the goal was to demonstrate, for the first time in literature to the best of our knowledge, the feasibility of the detection of grape bunches in images acquired in a generic vineyard, located in an unspecified geographical area with an unspecified grape variety.
Finally, regarding the assessment of different sizes of objects that kind of analysis was not relevant considering the available dataset. Indeed, all the images were acquired in frontal view at similar distances. There are not prospectic views of different sizes of similar objects and the different number of pixels of the objects are not due to different distances from the camera, but only to an actual size difference of those objects.
Language and style Several formatting errors can be found in manuscript (e.g. line 323)
Some of the terms are written with first capital letter, why? (precision, agriculture...).
Possible error line 68: lower than a threshold --> higher than a threshold?
Answer:
We thank you for the notice, we added the missing space.
We thank you for the notice, we have corrected all occurrences of terms mistakenly written with first capital letter.
We thank you for the error notice, we corrected the sentence since the local peaks lower than a threshold were not selected but they were eliminated (see line 96).
Round 2
Reviewer 1 Report
Thank you for the response.
Regarding the supplemented additions the manuscript could be accepted for publication.
Reviewer 2 Report
Dear authors,
Thank you for the answers you formulated on the comments. I consider them as sufficient and I can recommend the paper to be accepted.
Reviewer 3 Report
I still could not identify the exact novelty of the paper. It is not clear what new knowledge has been introduced here. As far as I know, the system is a made up of individual systems that have been proposed before and used in other applications. So what is the novelty of the paper?
Region proposal and Mask RCNN are heavily used for object detection. But why Mask RCNN among so many other systems? It is definitely not the most effective these days.
In addition, the algorithm for object detection is not really Mask RCNN but just a variant of RCNN likely to be Fast RCNN. It is misleading to call the algorithm Mask RCNN while you do not actually use it.
In field automatic detection of bunches under totally uncontrolled environment, the issue of illumination variation is very important especially in “uncontrolled environment” where the images are captured under various lighting conditions. From the authors response, it seems that they have shifted this matter and put them aside. For example, please see the work of “Computational deep intelligence vision sensing for nutrient content estimation in agricultural automation,” IEEE Trans. Automation Science and Engineering, 2018 “Regularized neural networks fusion and genetic algorithm based on-field nitrogen status estimation of wheat plants,” IEEE Trans. Industrial Informatics, 2016. I have previously pointed these papers out so that the authors could refer to it. I really could not agree that the important issue of illumination has been omitted in the work. It is methodologically not correct.
The authors responded that “We were able to gather thousands of images collected all around many vineyards in Sardinia (Italy). Some contained the entire vineyard, others in perspective the space between two rows or an entire row imaged from one end. The purpose of our work was to train a detector able to analyze images automatically acquired by a vehicle moving between the vine rows. Therefore we only selected photos acquired between the rows at a distance of about one meter from the leaf wall.”
From above, it is not clear how many samples of images have been collected. What is the dimension of each image? How many samples are used for training and validation? Why does it have to be one meter from leaf wall? What happens to the precision of detection when the distance is violated since this is supposed to be uncontrolled environment where the distance could vary?
It seems that mPA is only calculated at three points. Why just limit to this number?
Reviewer 4 Report
Authors tried to address all the points that I raised in my previous review. However, I still found that the manuscript is not written at satisfactory level.
- The results section is still weak, missing comparison with the strong baseline on these two datasets that are authors using.
- Authors tried to explain their motivation in theri answer (or informally contribution): "Here the goal was to demonstrate, for the first time in literature to the best of our knowledge, the feasibility of the detection of grape bunches in images acquired in a generic vineyard, located in an unspecified geographical area with an unspecified grape variety." I cannot agree with such statement because such approaches exist in literature [1]. In my last review I tried to point out authors what I could accept as a certain contribution - better results than a strong baseline and in-depth analysis of the used approach, pointing out its weaknesses and advantages.
- In same spirit as previous point, I cannot agree that labels for at least GrapeCS-ML dataset cannot be published. This dataset is freely available for download.
- Related work is still weak, missing important information.
- Some important references are missing (e.g like [2]).
- Again there are some errors in the text, eg. MAP (it shoud be mAP)
[1] Deep Neural Networks and Precision Agriculture for Grape Yield Estimation
[2] GBCNet: In-Field Grape Berries Counting for Yield Estimation by Dilated CNNs